# Robustness of Reinforced Concrete Frames against Blast-Induced Progressive Collapse

**Mattia Francioli [1], Francesco Petrini [1,\*], Pierluigi Olmati [2] and Franco Bontempi [1]**

[1]   Department of Structural and Geotechnical Engineering, Sapienza University of Rome, 00185 Rome, Italy; mattia.francioli@uniroma1.it (M.F.); franco.bontempi@uniroma1.it (F.B.)
[2]   Consultant, Explosion and Impact Engineering, Tokyo 153-8515, Japan; pierluigi.olmati@gmail.com
\*   Correspondence: francesco.petrini@uniroma1.it; Tel.: +39-06-4458-5072

**Abstract:** A quantitative procedure for the robustness and progressive collapse assessment of reinforced concrete (RC) frames under blast load scenarios is presented. This procedure is supported by multilevel numerical models, including nonlinear numerical analyses of the structural response of both local (i.e., response of the single structural element to the blast load) and global levels (i.e., response of the structural system to the blast-induced damage). Furthermore, the procedure is applied to a 2D RC frame structure. The novelty of the proposed procedure is that the global robustness is evaluated by the so-called "damage-presumption approach" where the considered damages are defined both in typology and extension depending on the blast scenario occurring at the local level. The dedicated local response analysis of a specified blast scenario leads to the proper definition of the so-called "blast-scenario dependent robustness curves".

**Keywords:** performance analysis; structural robustness; blast-induced damage; nonlinear dynamics

## 1. Introduction

Although the events of progressive collapse have a very low probability of occurrence, the consequences usually have a very high impact on society [1]. Progressive collapse can be triggered by many factors such as blast loading from explosives or gas leakage, design errors, vehicle impact, construction errors, debris impact, and other extreme loadings such as fire and earthquake [2,3].

In many instances, a significant propagation of direct damage to key structural components throughout the structure have produced a progressive collapse of residential, iconic, and public buildings, resulting in huge losses of life and property [4]. The interest in blast-induced damage started after an important event, which was the partial collapse of the Ronan Point tower in the UK in 1968 [5].

In this context it is important to introduce robustness as a crucial structural performance requirement. However, robustness is still the subject of controversy over its definition and quantification, and research activity on the topic has furnished substantial and useful recommendations for its assessment and for a robustness-oriented structural design [6]. A comprehensive definition of structural robustness is reported in Eurocode 1 (EN 1991-1-7, 2006) [7] as "the ability of a structure to resist events such as fires, explosions, impacts or the consequences of human error, without being damaged in a disproportionate way compared to the original cause", that explicitly refers to the kind of actions that are relevant to the robustness. Despite this mention of different hazards that the structural robustness should face, one of the most established procedures for robustness analyses in research is based on the so-called "damage-presumption approach", that is a "hazard-independent" analysis where a certain damage level is assumed for the structure, and the residual strength of the structure is then evaluated. Typically for a framed structure the presumed damage consists of the sudden removal of a column, something that will be referred to a "column removal" method in what follows. Common damage-presumption

approaches do not directly and univocally link the presumed damage to a specific hazard, something matching with the idea of interpreting the robustness as a structurally intrinsic characteristic [6–8]. This approach is not able to exhaustively link the hazard intensity measure parameters with the robustness performances as currently required by modern Performance-Based Design (PBD) approaches [9].

In addition, if the concept of robustness is well assimilated by the various regulations, one thing is still missing in the literature: an exhaustive definition of guidelines on structural design regarding robustness for specific hazards is necessary to develop the possibility of a common perception about this topic, in order to have the same PBD development as occurred in seismic engineering [3,10].

The goal of the paper is to contribute to filling the above-mentioned literature gap by proposing a procedure for the robustness quantification of RC frames under blast load scenarios. The procedure is supported by a multilevel numerical analysis, in the sense that it implies the nonlinear numerical analysis of the structural response both at the local level (i.e., response of the single structural element to the blast load) and at the global one (i.e., response of the structural system/frame to the blast-induced damage).

The novelty of the proposed procedure is that it provides a set of so-called "blast-scenario-dependent robustness curves" representing the robustness of the structure in relation to the damages that are coherently linked to the blast intensity measures. In other words, the robustness is evaluated by a "damage-presumption approach" where the considered damages are defined in a dedicated local response analysis and linked to a specified blast intensity.

## 2. Global Robustness of RC Frames under Column Removal Scenario

### 2.1. Structural Behavior Aspects

As reported in Starrosek [1], redundancy or compartmentation are the two main conceptual design strategies at global structural scale that can be pursued to satisfy robustness requirements, along with local ductility requirements. Modern fame structures are highly hyperstatic; therefore, they allow for alternative load paths and many local damages generally occur before a global collapse takes place. Furthermore, in the case of seismic-resistant structures, elements' cross sections are designed to have ductility and a dominant flexural mechanism at failure, something that is crucial for the development of some local deformation capacity that allows the global robustness. Therefore, the criteria of seismic engineering have a beneficial impact regarding the robustness of buildings [11]. Additional specific structural behaviors, like the membrane or catenary effects, lead to an overstrength which can be well exploited in case of damage, such as the removal of a key element like a column [12,13].

The nonlinear dynamic procedure for progressive collapse is the most thorough method of analysis in which a primary load-bearing structural element (e.g., a column) is removed dynamically and the nonlinear behavior of structural materials is taken into account. This allows larger deformations and energy dissipation through material yielding, cracking, and fracture.

However, as previously stated, when the structural robustness of frames is evaluated by the column removal method, if the column damage it is not linked with hazard-specific parameters, the assessment is not specifically representative of any kind of hazard.

### 2.2. Numerical Analysis for Structural Robustness

Regarding the residual load-bearing resistance corresponding to a column removal scenario several experimental studies have been conducted on reduced scale specimens mostly in order to study the progressive collapse resistance [14–16]. The main drawback of the experimental studies is that the column is usually removed under quasistatic loading conditions, thereby not capturing the dynamic effects of a sudden column loss, something that can play a prominent role in this kind of problem. From a numerical analysis point of

view, a material and geometric nonlinear finite element (FE) analysis must be used during the design and assessment of RC structures for robustness.

The nonlinear dynamic procedure for RC frames consists of analyzing the frame dynamic response under the sudden removal of a number "n" of columns for the frame starting from the static equilibrium configuration reached by the structure under vertical loads (generally due to the seismic "permanent + 0.3× variable" mass combination). The outcome of the nonlinear dynamic analysis can be of two typologies [17]: (a) after an initial damped transitory phase, the structure reaches a static equilibrium condition characterized by some residual plastic displacements; and (b) the collapse occurs.

Regarding the collapse, it can be defined to occur when: (a) there is "run-away" behavior (a structural behavior in which the time response or the load response diagrams are unconfined by certain boundary limits [18]) observed in the vertical displacement time history of the nodes around the removed column, or (b) the vertical relative drift between the beam–column nodes ($D_V$) located around the removed column reaches the value of 15–20% [19]. The latter is calculated starting from the vertical displacement of the node at the top of the removed column $\delta_V$ and the length of the beam to which the node belongs $L_b$:

$$D_V = \tan^{-1}(\delta_V / L_b), \tag{1}$$

If the outcome of the nonlinear dynamic analysis is not the collapse, an incremental static nonlinear analysis of the structure is carried out under lateral forces (pushover) in order to evaluate the residual capacity of the damaged structure [20]. If common approaches use the pushdown analysis to assess the residual capacity of the structure under gravity loads, the reason for using pushover instead is that it can be more significant in the case of multihazard scenarios/studies, which have become increasingly important during recent years in structural design. This is particularly significant for earthquake-induced explosions, where the evaluation of the residual capacity of the blast-damaged structures under earthquake aftershocks is crucial.

In this way, each number of simultaneously removed "n" columns is associated to a residual lateral force capacity ($\lambda u$) as evaluated by the pushover and expressed as a percentage ($\lambda u/\lambda\%$) of the force capacity ($\lambda$) of the nondamaged structure, evaluated by a similar pushover analysis.

Generally, the response to the initial dynamic analysis (typically represented by the time history of the vertical displacement of the node) is strongly influenced by several parameters regarding the analysis procedure or the structural model. One of these parameters is the removal time interval ($\Delta t_d$) for the column [21]: the less the $\Delta t_d$ for the complete removal of the columns, the more severe the consequent structural response. In this view, the identification or setting of the column removal time interval $\Delta t_d$ for a certain "n" would be of value.

*2.3. Robustness Curves*

As a result of the aforementioned numerical outcome, the robustness of the structure can be quantified and efficiently represented by the so-called "robustness curves" as introduced by Olmati et al. [17]. Robustness curves are represented on a Cartesian plane in which on the *x*-axis there is the damage level suffered by the structure ("n" in the previous section), while on the *y*-axis the corresponding residual force capacity percentage ($\lambda u/\lambda\%$ in the previous section) is reported. See, for example, the qualitative representation of robustness curves represented in Figure 1, where different markers represent different locations for the presumed damage along the structure. The steepness of the robustness curve when the local damage level is incremented is proportional to the decay of the residual capacity of the structure. In other words, starting from a certain damage level (and a certain residual capacity), the greater is the steepness of the robustness curve when the damage increases, the larger is the decrement of residual capacity suffered by the structure due to the damage increment.

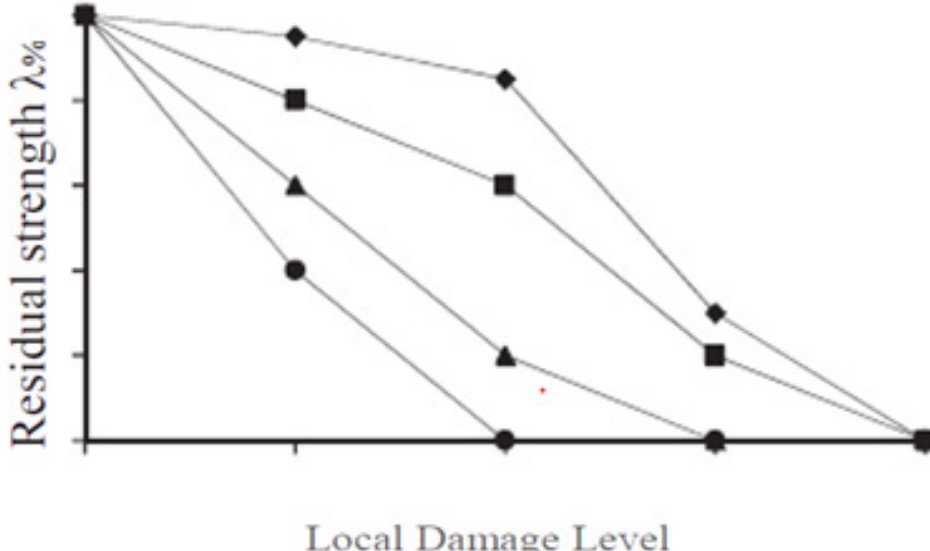

**Figure 1.** Typical robustness curves; different markers represent different locations for the presumed damage along the structure.

The procedure for the evaluation of the robustness curves in Figure 1 is depicted in the flowchart presented in Figure 2, already applied by the authors in Olmati et al. [17]. After a number $N_L$ of relevant locations along the structure have been individuated for the damage presumption (column removal), a set of damage scenarios implying an increasing damage level are defined and named "D-scenario (i,j)" (where "i" indicates the location and "j" the presumed damage level). Each D-scenario is analyzed starting from the lower damage level by implementing a nonlinear dynamic analysis (NDA in the flowchart). After each NDA, if the collapse does not occur, the pushover nonlinear analysis under lateral load is carried out to determine a point (residual capacity $\lambda u/\lambda\%$) of the robustness curve, and then the damage level is increased, and the NDA is repeated until the progressive collapse (as appropriately defined) occurs. In the proposed method, the progressive collapse of the structure is declared when the failure of a column adjacent to the columns removed or damaged due to the explosion is observed during the nonlinear dynamic analysis NDA. The procedure is carried out for different locations to obtain a set of robustness curves under blast presumed damage scenarios.

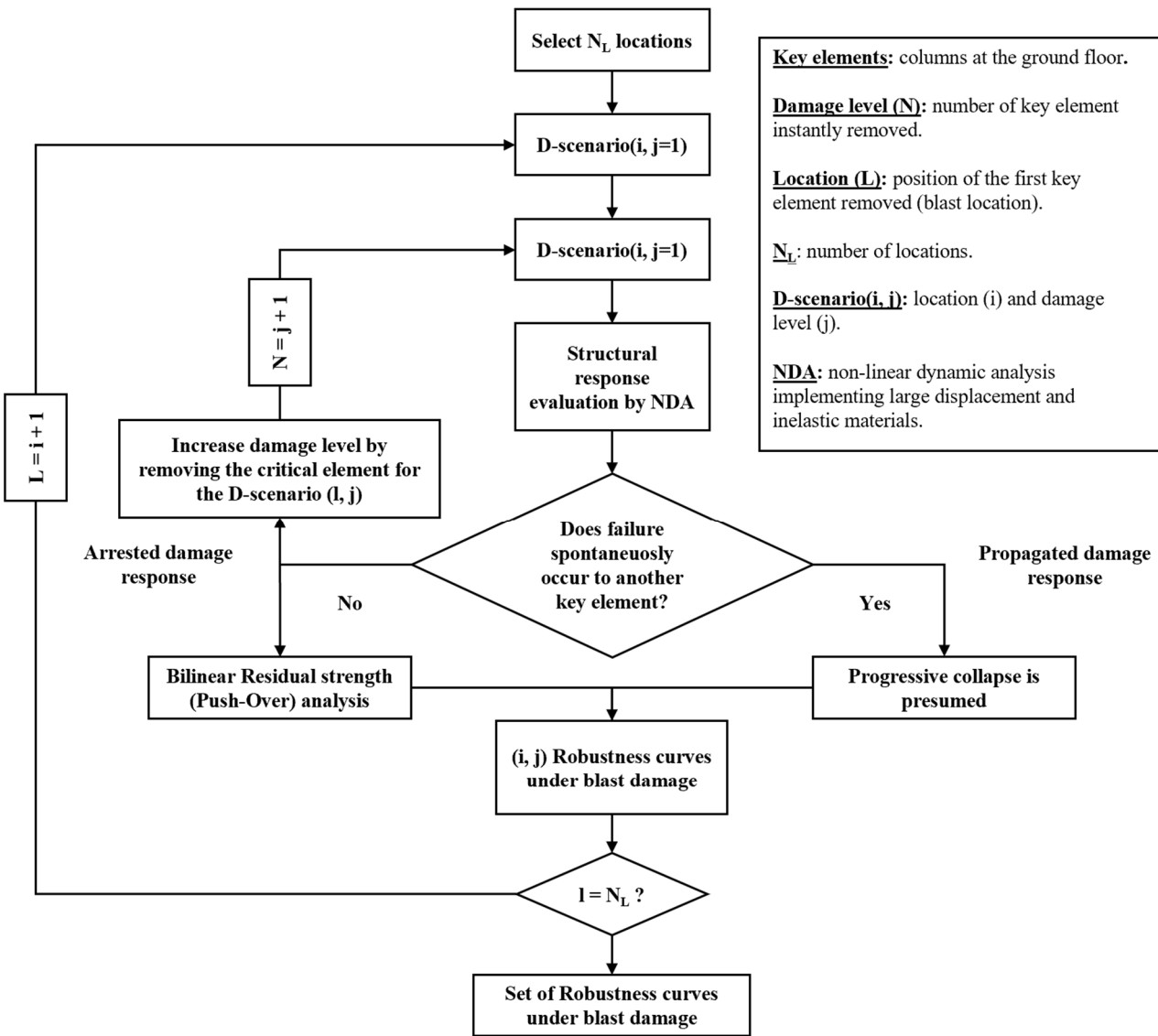

**Figure 2.** Flowchart of the procedure to evaluate the structural robustness against blast damage [17].

### 3. Blast-Induced Local Damage for RC Columns

*3.1. Blast Load*

A typical blast pressure–time profile at a fixed point in space for a blast wave in free air is shown in Figure 3. As is well known, the explosion initially generates an expansion of the air due to the release of energy; once it reaches a time $t_a + t_0$, the pressure changes in size. The absolute value of the peak pressure in the negative phase is typically smaller than the one in the positive phase. The area underpinned by the curve in the positive phase is the impulse of the blast, which has been proven to be a primary intensity measure for detonations [22]. In this study the blast intensity is determined by the equivalent TNT kilograms of explosive and the stand-off distance of the explosion from the element.

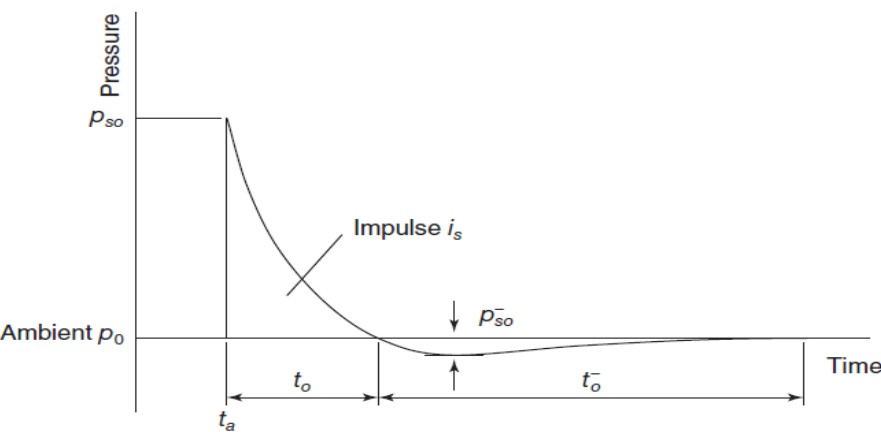

**Figure 3.** Typical blast pressure–time profile.

### 3.2. Local Models and Analyses for Blast-Damage Assessment

This section is about the nonlinear models implemented for investigating the performances of single column elements subjected to the blast pressure. In the view of the proposed framework, these are referred to as "local models" because they regard the local response of the structural portions of the building located around the explosion zone.

Furthermore, local CAPACITY and local DEMAND are defined here as:

- The CAPACITY is expressed in terms of lateral displacement thresholds and associated damage levels.
- The DEMAND is expressed by the peak lateral displacement and by the occurrence time of the peak lateral displacement counted from the explosion instant, called "peak response instant". This would appear to be unusual, but our reasons will be explained in Section 4.

With those demand and capacity definitions, the local models are used to evaluate the nonlinear static and dynamic responses of beam elements under a combination of axial static load and lateral impulsive load; and specifically, they are solved by:

(a) A static nonlinear (pushover) analysis to evaluate the local CAPACITY of the element under the lateral induced deformation typical of blast-loaded columns;
(b) Under a certain blast load intensity, a transient dynamic nonlinear analysis to evaluate the local DEMAND.

The typical result of a local numerical analysis for capacity assessment is shown in Figure 4. In this case the structural scheme adopted for the analysis is the one shown in Figure 5, with the "λP" load being statically incremented (by incrementally increasing values of "λ") as in the ordinary pushover analysis. Referring to the Italian Standards for the structural materials [23], the increasing of the concrete elastic modulus (Ec) and of the concrete strength (fc) under impulsive load has been taken into account by assuming Ec = 42,510 MPa and fc=41.73 MPa for a C28/35 concrete [24]. The steel rebars elastic modulus Es is taken as equal to 210,000 MPa, while the yielding stress fy (increased under impulsive load) is 510 MPa (B450C steel grade) [24]. Looking at the curve, it is possible to establish the threshold values for the peak displacements that are associated with different damage levels. In the examined case, there is an initial drop in the capacity curve corresponding approximately to the 0.5 of the total capacity at lateral displacement of 10 mm, while the second drop in capacity (ultimate displacement) is associated with the collapse of the column occurring with a lateral displacement of 30 mm correlated to a local damage equal to 1. It is worth noting that, an alternative to conducting a local capacity analysis, reference can be made to the capacity limits specified in the literature for RC columns (see, for example, the PDC-TR 06-08 document [25]).

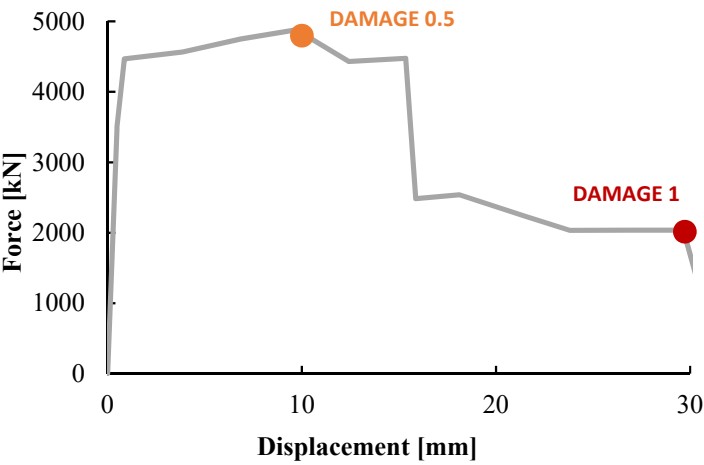

**Figure 4.** Capacity curve of the local element (RC column shown in Figure 5).

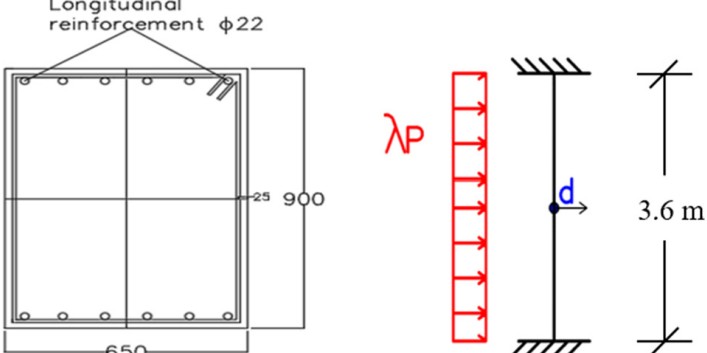

**Figure 5.** Cross section of the RC column analyzed; structural scheme considered and acting load. Sizes are in mm if not specified otherwise.

The typical results of a local numerical analysis for demand assessment conducted for different blast intensities is shown in Figure 6 (time histories of the midspan node lateral displacement of an RC column subjected to a lateral blast load). Results shown in Figure 6 are obtained for the RC column with the cross section and the loading scheme (uniform blast load) represented in Figure 5 with the "λP" load having the pressure–time profile shown in Figure 7 and corresponding to the different blast intensities (i.e., stand-off distance + equivalent TNT kilograms). These curves are obtained with the approximation of the typical blast pressure–time profile [26] taken from the formula of Mills:

$$P(t) = P_r(1 - t/t_d)e^{-\beta t/t_d}, \tag{2}$$

with

$$P_r = 2P_{S0}(7P_{atm} + 4P_{S0})/(7P_{atm} + P_{S0}), \tag{3}$$

$$P_{S0} = 1.772(1/Z^3) - 0.114(1/Z^2) + 0.108(1/Z), \tag{4}$$

$$i_{S0} = 300[0.5(W)^{1/3}], \; Z = R/(W)^{1/3}, \; t_d = 2i_{S0}/P_{S0}, \tag{5}$$

where R is the stand-off distance from the detonation point, W is the equivalent kg of TNT, $P_{atm}$ is the atmospheric pressure, and β is the decay coefficient, taken equal to 1.8.

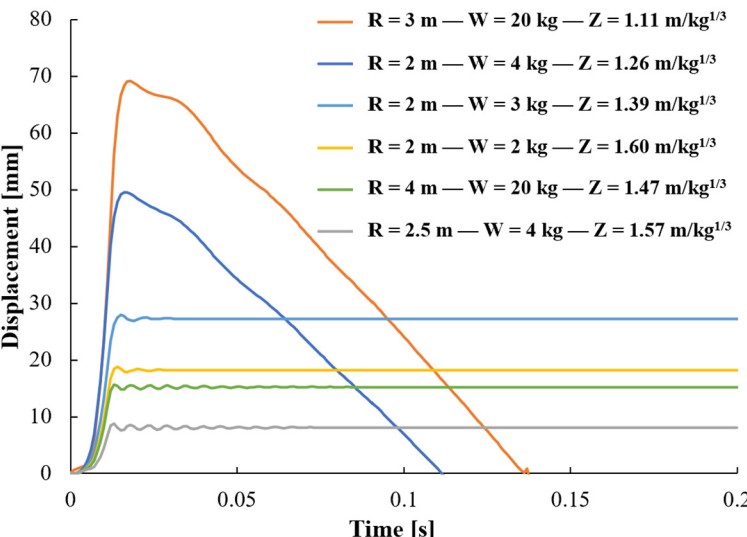

**Figure 6.** Typical results of a local numerical analysis for demand assessment: time histories of midspan node lateral displacement due to lateral blast load.

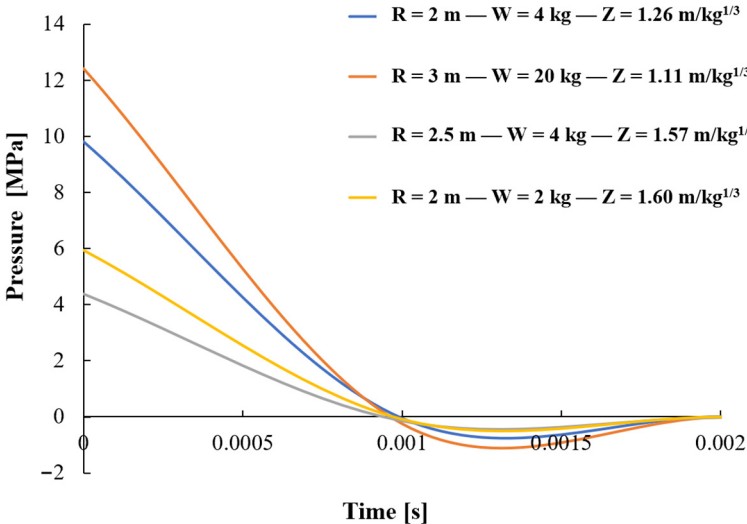

**Figure 7.** Typical blast pressure–time profile.

Different curves in Figure 6 represent the response obtained for different blast intensities, which are identified, as previously stated, by the parameters: blast stand-off distance R (meters) and equivalent kilograms of TNT W (kg). In the figure, the scaled distance Z corresponding to each intensity is also indicated. Numerical analyses are carried out by the SAP 2000® commercial structural code [27], by using beam finite elements (FEs) for the column and by implementing large displacement solutions and the plastic hinges approximation for modeling material nonlinearity, the latter being modeled by considering the bending behavior at the two ends and at midspan. The elastic–plastic bilinear hardening model has been implemented for the hinges, with a rotational ductility ratio $\theta_{ult}/\theta_y$ ($\theta_{ult}$ and $\theta_y$ being the ultimate and the yielding rotation, respectively) fixed to 30 and the "drop to zero" option switched on in SAP2000®. As can be seen from the figure, different blast intensities lead to different structural responses: the intensities (2 m-2 kg), (2 m-3 kg), and (2.5 m-4 kg) lead to a "damaged response" for the column, with some residual displacements after the transitory response, while the intensities (2 m-4 kg) and (3 m-20 kg) lead to the failure of the elements, something which can be recognized from the value of the maximum displacement reached (which is larger than the 30 mm limit associated with the damage 1 in Figure 4), and from the consequent decreasing of

the displacements toward zero, something that is unrealistic and numerically induced. From Figure 6, it is observed that the structural response is more sensitive to the stand-off distance than to the equivalent TNT kilograms. In fact, focusing on the cases with 2 m stand-off distance, the TNT kilograms must reach the value of 4 kg to lead the collapse (2 kg and 3 kg lead to some damaged response with residual displacements), while if the stand-off distance rises to 3 m (+50%) we need to increase the TNT value to 20 kg (+400%) for the collapse.

The final outcome of the local demand analysis is the demand function shown in Figure 8. The representation of the maximum displacement at midspan related to the time at which the element reaches its peak displacement $\Delta t_p$ (called "peak response instant" in the following sections) for each scenario has a key role in the definition of the local blast demand for the column and the induced damage.

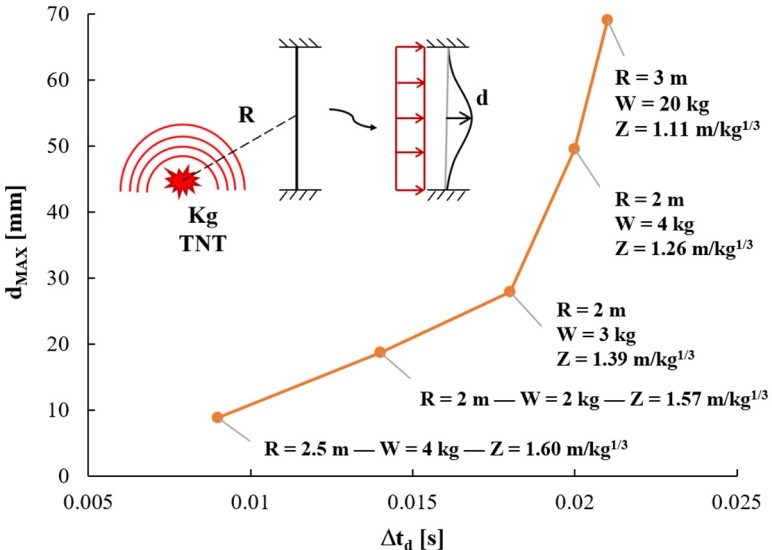

**Figure 8.** Blast local demand curve midspan maximum displacement vs. peak response instant.

According to data in Figures 5 and 8, it is now possible to associate to the damage of the element, the stand-off distance and equivalent kilograms of TNT that produce a specific damage level to the column. In this case, damage 1 occurs when 3 kg of equivalent TNT detonate 2 m away from the element (the $d_{MAX}$ from Figures 6 and 8 is equal to 30 mm).

## 4. Application to an Existing Structure

### 4.1. Case Study Structure and FEM Model

In order to apply a multiscale procedure for the robustness quantification of RC frames under blast load scenarios, a 2D RC frame structure is considered (Figure 9). The building to which the 2D frame belongs is part of a very complex hospital system completed in the early 2000s. It is an RC structure made of concrete C28/35 and steel B450C under the classification of Italian Standards [23]. The structure is modeled using the SAP 2000® structural code, by defining the nonlinear properties of the materials. The nonlinear behavior is implemented using the approximation of plastic hinges, which are obtained from the moment-rotation relationship (M-θ) evaluated from the equations provided by the Italian Standards NTC2018 [23]. All the columns of the ground floor have the cross-section already presented in Figure 5. Moreover, geometric nonlinearity is considered with large displacement and P-Δ options. As stated previously, the 2D frame is extracted from a complex structure and in order to simulate the contribution to the catenary effect and the membrane effect provided by out-of-plane beams and by the slab respectively, a dedicated nonlinear beam finite element is added, connected in parallel to each beam of the 2D frame, and named "special element" in what follows. This latter element has been modeled as an ordinary beam element provided with axial and bending plastic hinges connected to

the columns of the 2D frame. The special elements have been calibrated on the basis of the membrane behavior of the floor slab as follows: (i) a dedicated fiber-based plastic model has been used in order to identify the axial and bending behavior of the 3D floor module (in-plane and out-of-plane beams plus slab); (ii) then, the plastic hinges and the elastic stiffness of the additional special elements have been calibrated in such a way that the in-plane beams of the 2D frame plus the special elements, were able to provide the same vertical elastoplastic strength and stiffness of the above mentioned floor fiber model, then taking into account for the out-of-plane membrane and catenary effects in the 2D frame model.

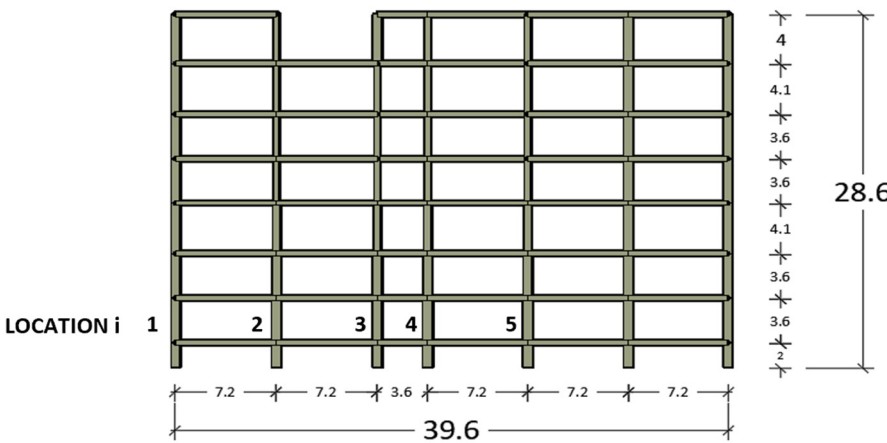

**Figure 9.** 2D RC frame structure and different locations assumed for the blast damage (sizes in m).

Considering the 2D frame structure, a nonlinear static analysis has been made in order to evaluate the effectiveness of these new elements. After the removal of a certain column, a pushdown analysis was conducted by amplifying vertical loads. The pushdown analysis was conducted on two models, whose difference was the presence of the aforementioned out-of-plane elements. During the analysis, the vertical displacement of the node at the top of the removed element was monitored, together with the resultant vertical forces. The outcome for the pushdown analysis carried out when column 5 was removed (see Figure 9) is shown in Figure 10; the presence of the membrane/catenary effect allows an increase in strength.

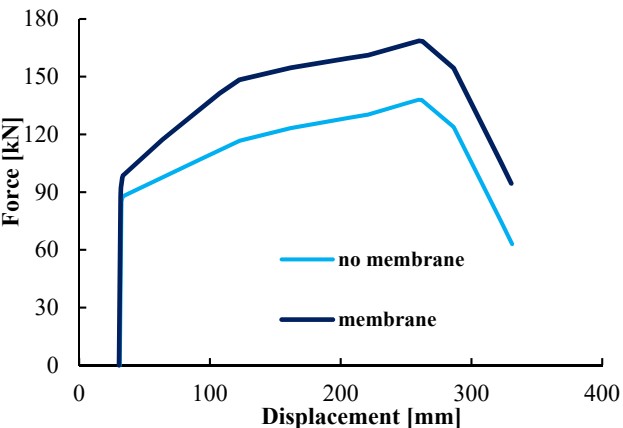

**Figure 10.** Membrane forces above column 5 in RC beam–column substructures under push-down analysis.

The two-level numerical analyses described in previous sections are applied to the case study frame in order to:

- Evaluate the global robustness curves of the structure; and

- Evaluate the blast local demand curve for the columns at the locations indicated in Figure 9.

The results obtained by the two analysis levels are joined together to evaluate the so-called "blast-scenario dependent robustness curves".

### 4.2. Global Robustness Results and Sensitivity Analysis

The procedure for the robustness assessment of RC frames under blast load scenarios (also presented in the flowchart of Figure 2) is described in this section.

After having defined a certain number of locations $N_L$ (column 3 and column 5, see Figure 9), analyses for each of the D(i,j)-scenarios (i = location, j = damage level) are developed (in order to evaluate the global robustness of the structure). A particular location is considered and starting from a certain damage level the structural response is evaluated by the nonlinear dynamic analysis. Here the damage level is intended as the number of columns that are removed in the global analysis. If failure doesn't occur spontaneously (as defined in Section 2.2) the typical structural response is the one shown in Figure 11. Successively, the residual strength of the structure is identified using a nonlinear static analysis, then the damage level is increased (i.e., an additional element is removed together with the previous one) and, again, the structural response is evaluated. During the analysis that provides the evaluation of the residual strength of the structure (i.e., pushover analysis), the residual lateral force capacity ($\lambda u$) considered is the one that corresponds to the first occurring condition between the "run-away" behavior observed in the vertical displacement time history of the nodes around the removed column or the experimentation of a vertical drift ratio ($D_V$) bigger than 15% (see Section 2.2). Obviously, if the number of considered locations $N_L$ is different from 1, all is repeated $N_L$ times. Before starting to apply the procedure in order to identify the robustness of the structure under blast loads, various locations of hypothetical damage were assumed (Figure 9). Depending on which column is removed, the results obtained by the lateral pushover analysis are shown in Figure 12: the internal columns (No. 3, No. 4, and No. 5) are characterized by a bigger tributary area under vertical loads and the hypothetical damage to one of them could cause a bigger reduction in the capacity.

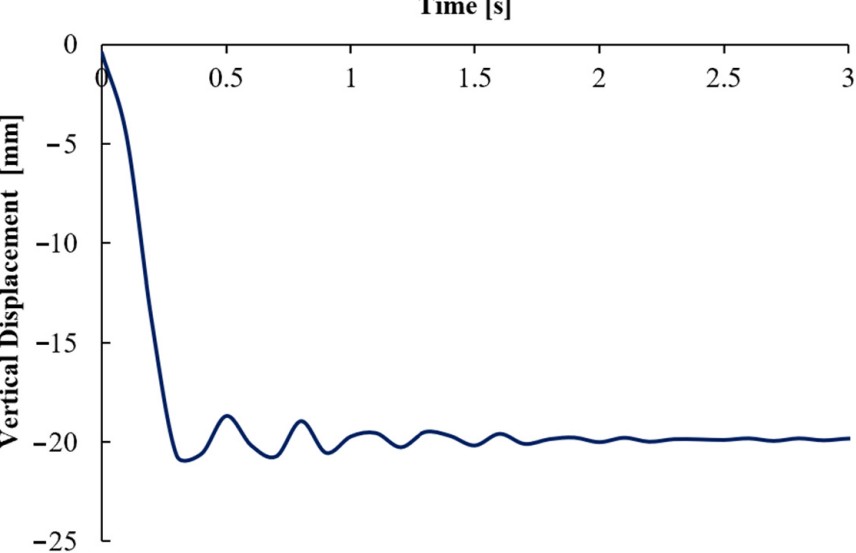

**Figure 11.** Typical displacement of the node at the top of column 3.

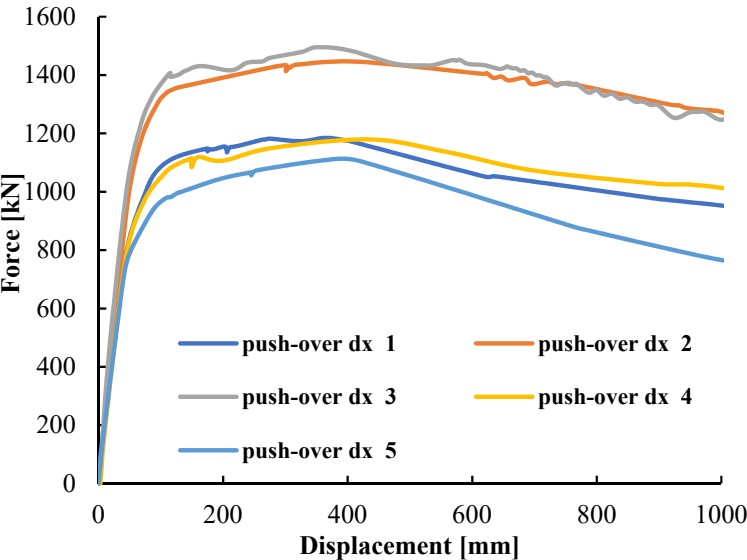

**Figure 12.** Pushover curves of the 2D frame structure due to different locations of damage.

The curves obtained by the pushover analysis can be bi-linearized. The curve obtained by analysis is replaced with a simplified curve which has at first a linear part and then a perfectly plastic plateau at the $F_Y$ value for the force. The slope of the linear part is identified imposing the passage for the point $0.6F_{MAX}$ of the original capacity curve, while the value of $F_Y$ is obtained by imposing the equality of the areas underpinned by the bilinear curve and by the capacity curve for a fixed maximum displacement $d_U$.

Once each location is established, some sensitivity analyses have been carried out by varying different parameters [28]: the damping ratio ($\zeta$) and the removal time interval of the column ($\Delta t_d$). The variation of each of these parameters influences the behavior of the structure in terms of ultimate strength and deformation; moreover, the $\Delta t_d$ is a very important parameter that allows the connection between the global level and local level analysis because the time interval of column removal $\Delta t_d$ can be considered as the time during which the damage propagates and affects the structural element under blast load effects (see Appendix A for a full report on the sensitivity analyses performed). The effect of different $\Delta t_d$ values on the structural robustness are shown in Figure 13, where the robustness curves obtained for $\Delta t_d = 0.01; 0.02; 0.3; 0.5$ s, and for the location 2 case, are compared with each other. As expected, the less the $\Delta t_d$, the lower the residual capacity obtained for a fixed damage level.

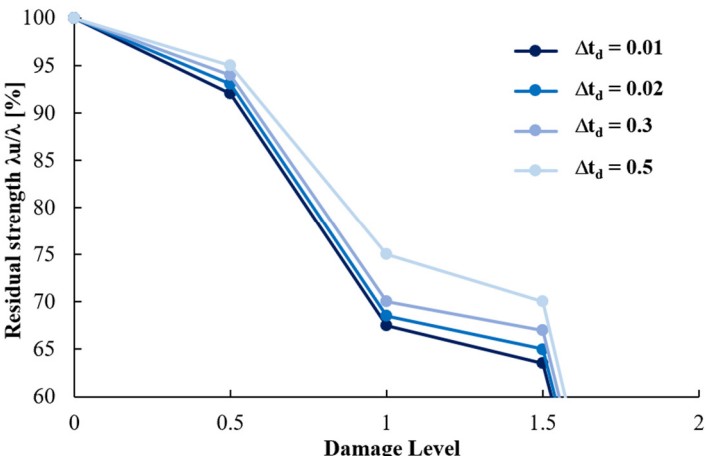

**Figure 13.** Robustness curves for 2D RC frame structure for location 2: effect of variation of $\Delta t_d$ (removal time of the column). $\Delta t_d$ values are expressed in seconds.

Figure 14 shows the results of the discussed analyses in terms of robustness curves, obtained as described in Section 2.3. Using the terminology presented in the flowchart of Figure 2, two locations are considered ($N_L = 2$): location 1 implies that the first element removed is column 3, while for location 2 the first element removed is column 5. In both cases, damage level 2 corresponds to the progressive collapse of the structure due to the spontaneous failure of a column adjacent to the removed or damaged one as a result of the explosion. Damage level 0.5, instead, corresponds to the loss of 50% of the transversal section; this means that the explosion results in a loss of element stiffness and capacity, but not in a collapse. The damaged element continues to carry the axial load but there are no dynamic effects due to the loss of the column. It should be noted that for damage level 1.5, considerations are similar to those reported for damage level 0.5: in this case, one column is completely removed, and the loss of 50% of the second column's section is considered. All the pushover curves are reported in Appendix A, where the effect of different values of the investigated parameters (damping $\zeta$ and removal time of the column $\Delta t_d$) is discussed.

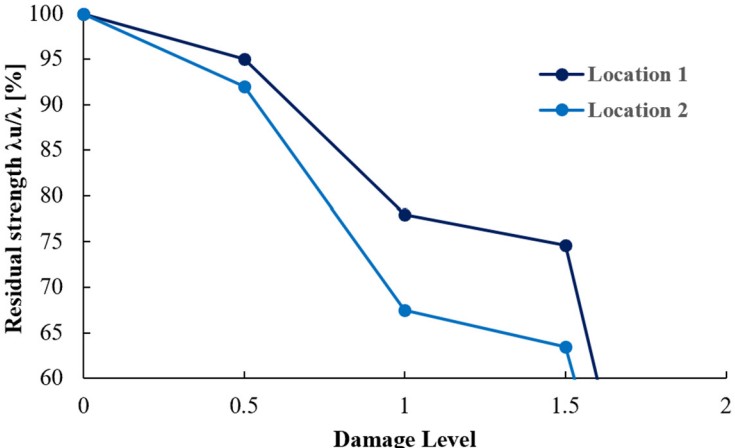

**Figure 14.** Robustness curves for 2D RC frame structure for location 1 and location 2.

*4.3. Local Demand*

As previous discussed, it is possible to associate the different values of $\Delta t_d$ to the different intensity measures of the explosion (stand-off distance from the ignition and equivalent kilograms of TNT). In this view, the second step of the procedure contemplates the local analyses from which the results already presented are obtained. In particular, the focus is on the evaluation of the capacity curve of the column (Figure 4) and on the assessment of the blast intensities, stand-off distance, and equivalent kilograms of TNT, which define the blast local demand (Figure 8). It is then essential to link together these two analysis levels: from the curve in Figure 4 it is possible to evaluate the capacity of the element and the value of displacement corresponding to a certain damage level (i.e., damage level 1, failure of the element, occurs in correspondence with the ultimate displacement of the midspan node; damage level 0.5, the partial failure of the element, occurs in correspondence with the first drop of the element capacity from its maximum value, see Figure 4); from the demand relationship between the intensity measures of the explosion and the peak response instant $\Delta t_p$ shown in Figure 8, it is possible to associate the specific value of the column removal time $\Delta t_d$ with a certain blast intensity. This can be done by interpreting the peak response instant $\Delta t_p$ obtained from the local analysis, as the column removal time $\Delta t_d$ used in the global analysis (i.e., $\Delta t_d = \Delta t_p$).

*4.4. Blast Scenario-Dependent Robustness*

The two different steps of the procedure for the robustness assessment of RC frames under blast load scenarios discussed in previous sections can be joined together to produce an innovative outcome that takes a first step towards the covering of the literature gap highlighted in the introduction of the paper: the so-called blast scenario-dependent

robustness (BSR) curves are proposed for correctly linking the structural robustness with the blast intensity.

The BSR curves are obtained point-by-point from the results presented in previous sections as synthetically shown in Figure 15 by completing these steps:

(a) LOCAL DAMAGE PRESUMPTION. First, by using the local capacity curve defined in Figure 4, a certain presumed local damage level is associated with a certain peak lateral displacement $d_{peak}$;

(b) BLAST SCENARIO DEFINITION. Second, by means of the blast local demand curve in Figure 8, it is possible to associate the $d_{peak}$ value previously identified with a particular blast scenario characterized by a certain blast intensity (a stand-off distance and a certain value of equivalent kilograms of TNT), and correspond it with a peak response instant $\Delta t_d$;

(c) ROBUSTNESS SELECTION. Finally, the appropriate robustness for the presumed local damage above can be selected among the robustness curves evaluated in Figure 13 as the one obtained by the column removal time interval equal to the peak response instant $\Delta t_d$ and then associated with the above-identified blast scenario.

Note that: if the presumed damage level is larger than 1, step (b) above should be performed by considering more than one column demand curve, with the stand-off distances from each column evaluated by assuming a specific location for the explosion.

By repeating the steps (a) to (c) for the different local damage levels the BSR curve is obtained. Those BSR curves have the residual capacity of the structure on the vertical axis associated not only with the presumed damage level but also with the corresponding blast scenario. Thus, such robustness curves are specifically related to the blast hazard.

The BSR curves obtained for the case study structure are shown in Figures 16 and 17 for location 1 and location 2, respectively. The value of the damping ratio is fixed to 4% for all the analyses.

Regarding the first location (the first removed element is column 3), as described in Figure 15a, a local damage level of 0.5, which corresponds to a loss of 50% of the transversal section of the element, can be associated with a lateral displacement of the midspan node of the column of about 15–20 mm, corresponding to a first loss of "local" capacity; similarly, a local damage level of 1 (complete failure of the column) can be associated with a lateral displacement of about 30 mm. Thus, it is possible to produce a blast-dependent robustness curve (Figure 15d) starting from the global robustness: after having associated the occurrence time interval $\Delta t_d$ with the considered displacement at midspan node (Figure 15b), the values of $\lambda_u / \lambda$ connected to the corresponding removal time interval of the column are selected from Figure 15c. In this case, for example, a blast scenario with a stand-off distance equal to 2 m and 3 equivalent kg of TNT, which is characterized by a peak response instant of about 0.02 s, is associated to the local damage 1; the value corresponding to damage level 1 of the robustness curve with $\Delta t_d = 0.02$ s is considered from Figure 15c and used for the construction of the BSR curve of Figure 15d at the same damage level. A local damage level of 0.5 occurs for a blast scenario characterized by a stand-off distance equal to 2.5 m and 4 equivalent kg of TNT (peak response instant equal to 0.01 s): the value of robustness belonging to curve $\Delta t_d = 0.01$ s is used in order to identify the point on the BSR curve of Figure 15d at a damage level of 0.5.

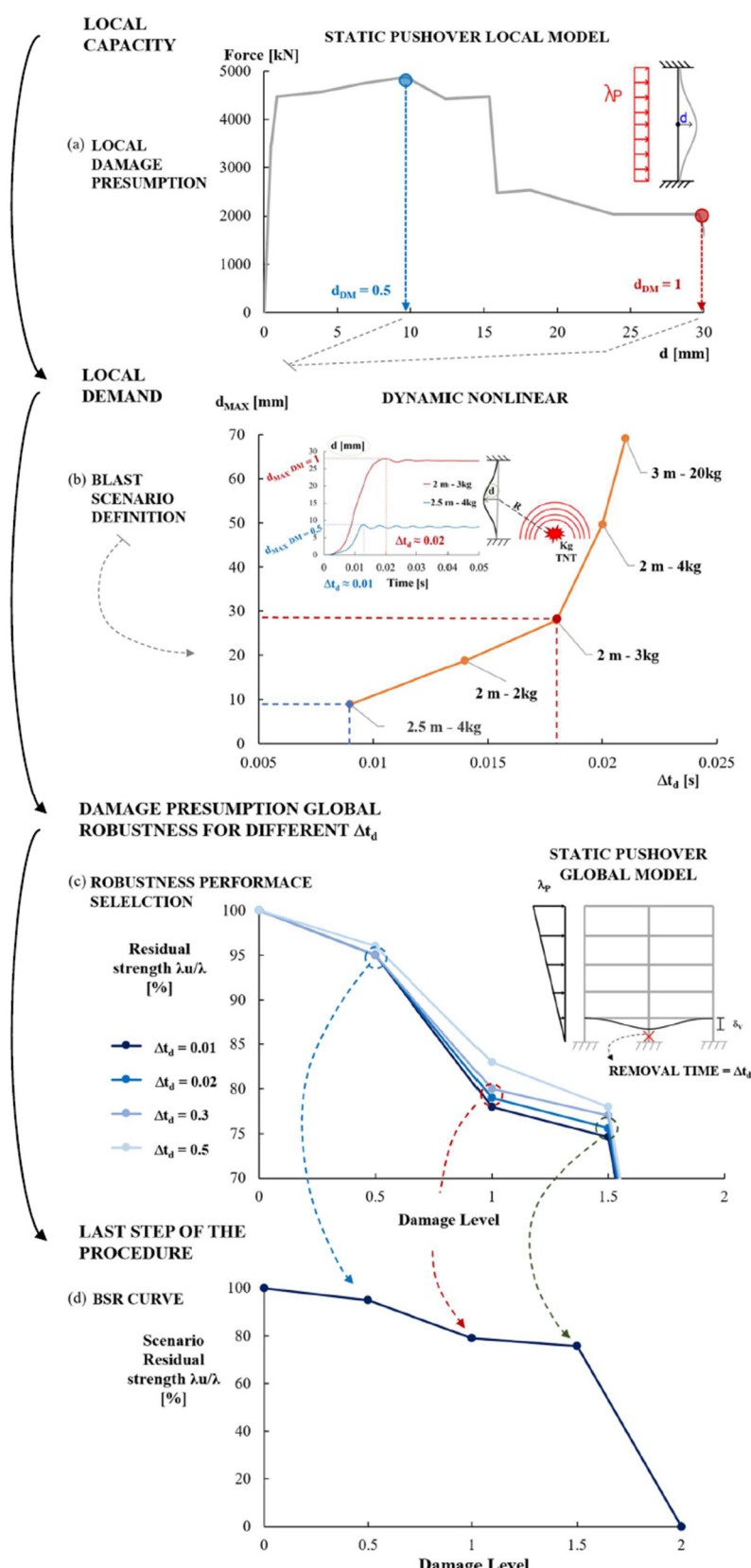

**Figure 15.** BSR curves evaluation summary. Local damage presumption (**a**); blast scenario definition (**b**); robustness performance selection (**c**); and BSR curve (**d**).

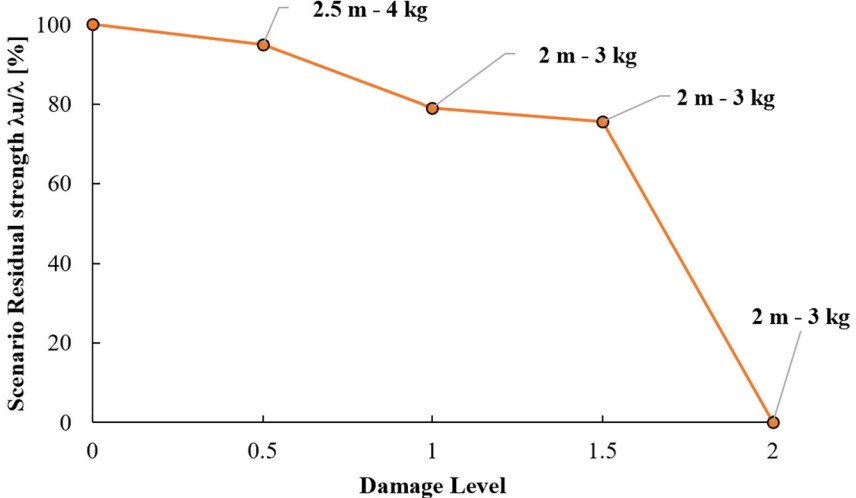

**Figure 16.** BSR curve for location 1.

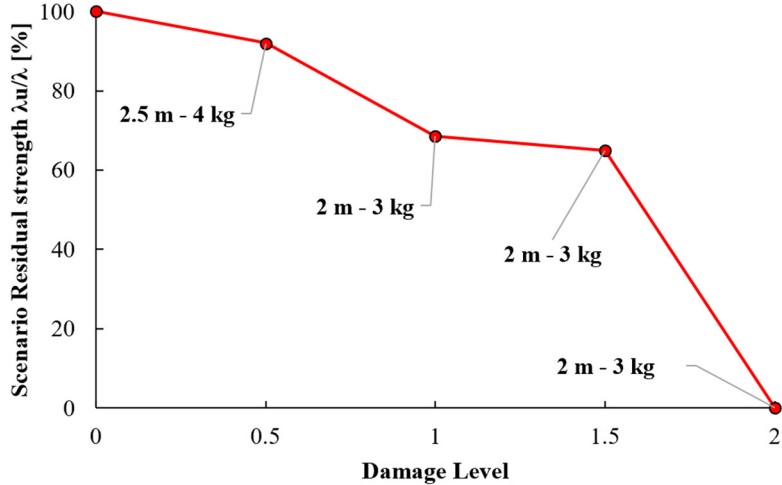

**Figure 17.** BSR curve for location 2.

## 5. Conclusions

A procedure for the robustness quantification of RC frames under blast load scenarios has been proposed where global and local structural behaviors are linked together. Therefore, the proposed procedure is called the "blast-scenario dependent robustness" (BSR) procedure.

This robustness quantification consists of two different steps of analysis: (1) the first level involves the investigation of the local structural behavior and allows the evaluation of the capacity and the demand of the single element which is subjected to explosions (here identified as "blast scenarios"); (2) the second level is connected to a global assessment of the structure and needs column removal analyses and pushover analyses to evaluate the behavior of the structure under sudden column removal scenarios first, and then post-damage behavior to find the residual capacity of the damaged structure.

The BSR curves constitute an advanced design tool for structural robustness under blast load, since they allow the evaluation of the structural robustness performances by referring to the load intensity, and then allow the correct evaluation of the robustness effect of both structural hardening measures and hazard mitigation measures.

Further developments for this procedure consist of considering uncertainties in the robustness evaluation. The aim of such development goes towards the probabilistic performance-based design or assessment. For example, in works like [29–31], the hazard and structural behavior are fully stochastically described but a connection with structural

robustness is missing. Therefore, the here proposed procedure could be improved by including a probabilistic framework to fill this gap, in a similar way to what has been suggested in [32].

**Author Contributions:** Formal analysis and original draft preparation, M.F.; conceptualization, methodology, supervision and original draft preparation, F.P.; review and editing, P.O.; supervision, review and editing, F.B. All authors have read and agreed to the published version of the manuscript.

**Funding:** This research received no external funding.

**Data Availability Statement:** The data are not publicly available due to privacy reasons.

**Conflicts of Interest:** The authors declare no conflict of interest.

## Appendix A

This appendix presents the complete results of the sensitivity analyses carried out, to understand the effects of the variation of the damping and of the removal time interval of the column ($\Delta t_d$). Considering the first parameter, Figure A1 shows the effects of the variation of the damping ratio $\zeta$ for location 1 when the removal interval $\Delta t_d$ is set equal to 0.02 s. As the damping index increases, the maximum vertical displacement and the time necessary to dampen the free oscillations of the removed column node decreases. For successive analyses, the damping ratio is set equal to 4% since smaller values (e.g., 1% in the figure) do not determine a significative difference in terms of "damping" time. For the sake of completeness, it has to show that a very large damping index, such as 0.5, leads to an almost absence of oscillation. The bi-linear pushover curves in Figure A2 show the case of damage 1, with the instantaneous removal of column 3, and depict the influence of the damping index parameter at the same removal time interval $\Delta t_d$, set equal to 0.02 s. As the damping index decreases, the above described dynamic amplification effect leads to a decreasing in both the stiffness and strength of the damaged frame (i.e., after the removal of the column) under lateral load.

Figures A3 and A4 show the effect of the variation of $\Delta t_d$ for location 3. As it appears in Figure A3, which shows the results of a column removal analysis that captures the effects of amplification in terms of displacement and in terms of geometric and material nonlinearities, the displacement of the node at the top of the removed column increases as $\Delta t_d$ decreases. There is a small difference between the cases with $\Delta t_d = 0.5$–0.3 and $\Delta t_d = 0.01$–0.02. It is also possible to note that the value of $\Delta t_d$ has a certain influence on the amplitude of the oscillations around the residual displacement and on the damping shown in the time histories.

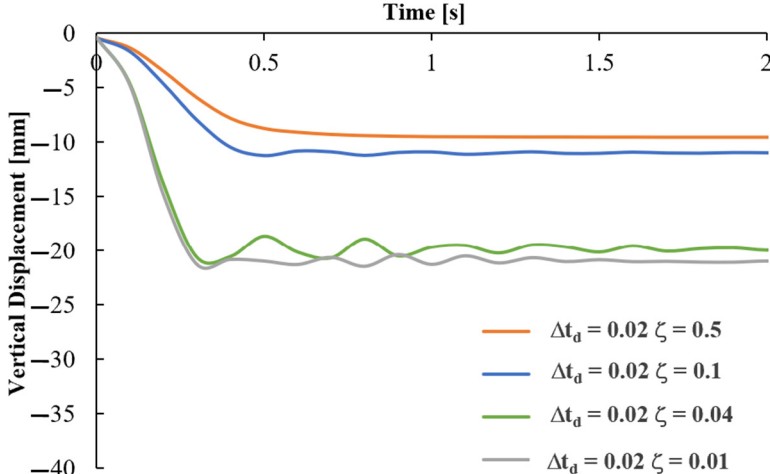

**Figure A1.** Displacement of the node at the top of column 3 (location 1): effect of variation of damping ratio ($\Delta t_d$ values are expressed in seconds).

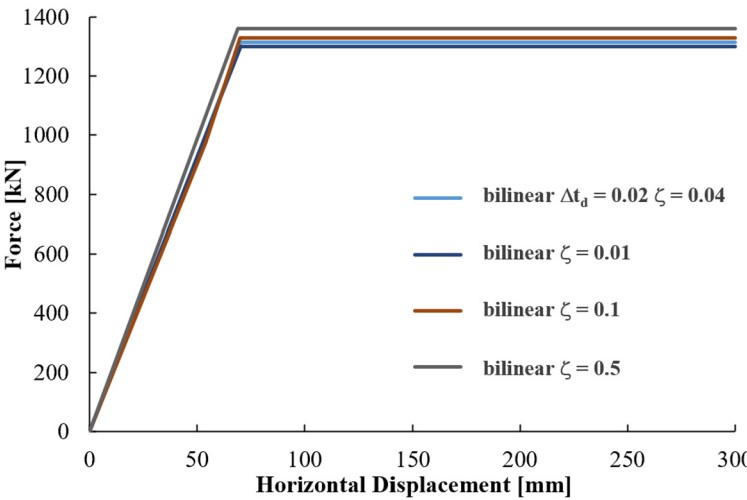

**Figure A2.** Pushover curve of 2D RC frame structure—damage level 1: effect of variation of damping ratio.

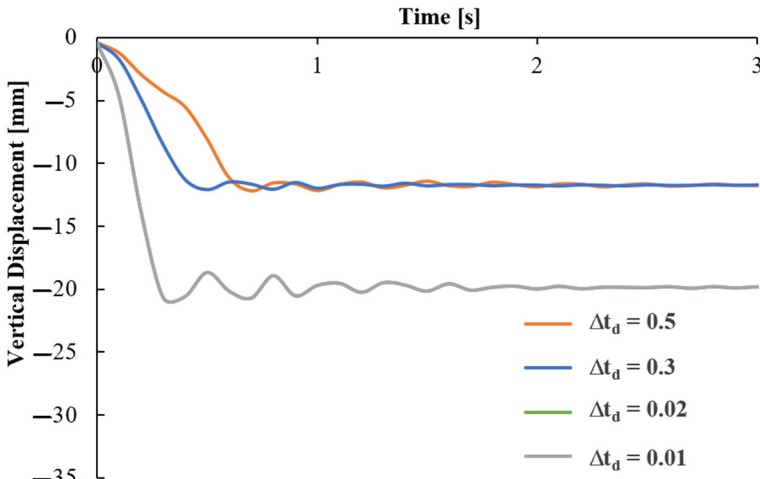

**Figure A3.** Displacement of the node at the top of column 3: effect of variation of $\Delta t_d$ (removal time of the column). $\Delta t_d$ values are expressed in seconds.

Figure A4 shows the results of the pushover analyses for location 1 with damage level equal to 1 (column 3 removed) and how the variation of $\Delta t_d$ affects the capacity of the damaged structure. Although there are just slight differences between the curves for the cases considered, if the $\Delta t_d$ value also decreases the overall capacity decreases. It is important to understand the effect that different values of $\Delta t_d$ have in terms of decreasing the capacity of the structure because this parameter can be used to simulate the damage induced by different blast scenarios.

In Figures A5 and A6 it is possible to notice the effect of the variation of $\Delta t_d$. Coherently with Figure A4, where the pushover curves for damage level 1 are reported, the decrease of $\Delta t_d$ determines a decrease in the residual capacity for both locations. Damage level 2 always determines the progressive collapse of the structure, while damage level 1 causes a drop in initial capacity of about 20% for L1 and about 30% for L2. Similarly, Figures A7 and A8 show the effect of the variation of the other parameter investigated, the damping ratio, with a defined removal time of the column ($\Delta t_d = 0.02$ s). The trend related to capacity losses remains the same as previously discussed: if the damping index increases, there is an increase in the capacity of the RC frame compared to cases with a smaller damping value. Even in this case, damage level 2 causes the collapse of the structure and damage level 1 determines a reduction of the structure's capacity.

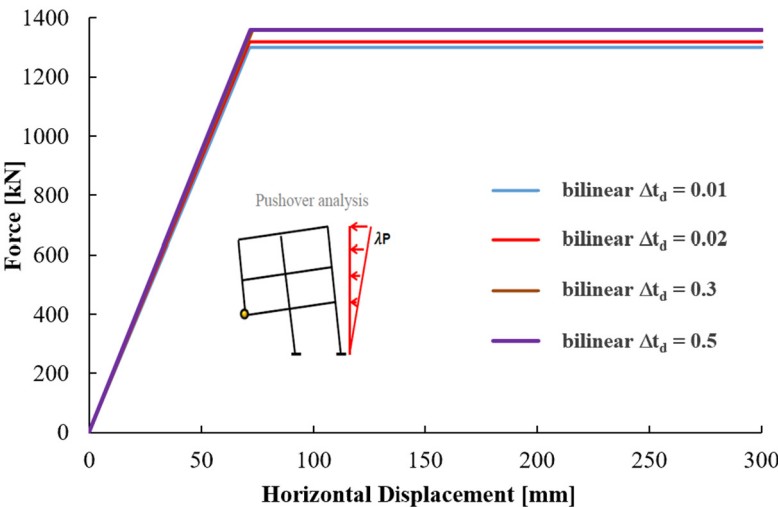

**Figure A4.** Pushover curve of 2D RC frame structure—damage level 1: effect of variation of $\Delta t_d$ (removal time of the column). $\Delta t_d$ values are expressed in seconds.

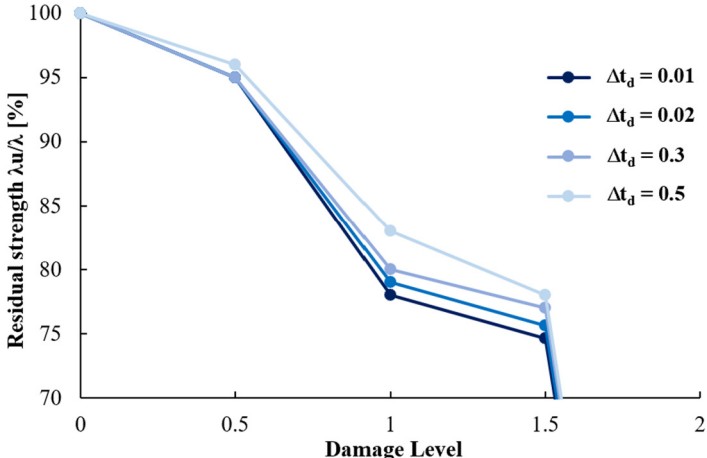

**Figure A5.** Robustness curves for 2D RC frame structure for location 1: effect of variation of $\Delta t_d$ (removal time of the column). $\Delta t_d$ values are expressed in seconds.

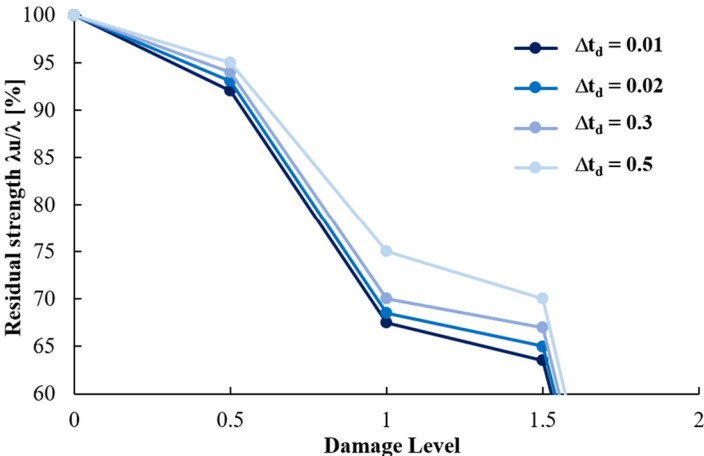

**Figure A6.** Robustness curves for 2D RC frame structure for location 2: effect of variation of $\Delta t_d$ (removal time of the column). $\Delta t_d$ values are expressed in seconds.

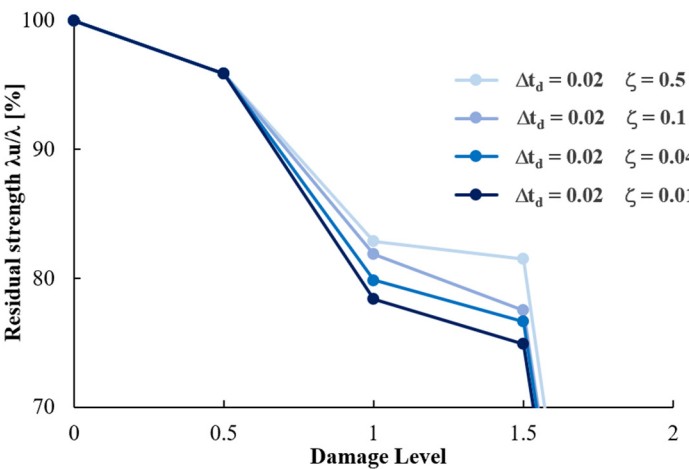

**Figure A7.** Robustness curves for 2D RC frame structure for location 1: effect of variation of damping ratio. $\Delta t_d$ values are expressed in seconds.

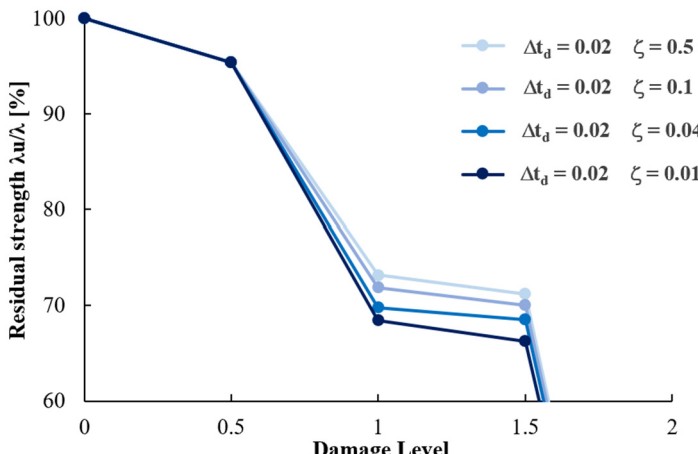

**Figure A8.** Robustness curves for 2D RC frame structure for location 2: effect of variation of damping ratio. $\Delta t_d$ values are expressed in seconds.

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
