# Peer review of "Robustness of Reinforced Concrete Frames against Blast-Induced Progressive Collapse"

_vibration, doi:10.3390/vibration4030040_

Round 1

Reviewer 1 Report

Dear Dr. Petrini:

This new methodology that you are presenting in this paper for evaluating the robustness of RC frames subjected to blast loads is quite interesting. The paper has a good structure, and you present your methodology fairly well. However, I have a number of comments for some features of your methodology that I explain in more detail in Technical Comments below. Also, the paper has several grammar and spelling issues that affect the readability of the paper. Below, on the Editorial Comments section, I provide a sample of these issues but there are more. I strongly recommend that you carefully proofread and revise the paper to further improve the quality of the language.

Technical Comments

General comments organized per section.

  1. Introduction

There is a lot of good content on this section. I would suggest adding references that will support the second sentence (lines 28-31) i.e., add a couple of references of notable examples of progressive collapse events.

  1. Global Robustness of RC Frames

2.1  Structural Behavior Aspects

  • Hyper static structures are in general most of the typical building structures, even if they are not designed for earthquakes. Therefore, the statement in line 80 is not valid. Perhaps ductility is a unique feature of seismic design as you mention in line 82.
  • Please add a reference for the nonlinear dynamic procedure that you refer to in line 88. UFC 4-03-023 can be a good reference.
  • You mention the column removal method in line 94 but you do not provide references that describe this method. The column removal method has certain requirements that are included in a number of design codes. Please add some or at least one as a reference.

2.2  Numerical Analysis for Structural Robustness

  • In line 101, you refer to a couple of older publications of experimental studies of progressive collapse resistance. There is more recent work on experimental studies for PC that you may consider adding as reference too.
  • What do you mean in line 114 when saying “run-away behavior”. Can you please explain? Shouldn’t the percentages on line 117 be 15-20% instead of 20-15%?
  • In lines 120 and onwards, you mention that after the removal of the column, a pushover analysis is carried out to evaluate the residual capacity of the damaged structure. Is there a specific code provision or some sort of reference that you base this statement on? It will make more sense if after the removal of one or more columns, to perform a pushdown analysis as well to assess the residual capacity of the structure under gravity loads. Lateral capacity loss is not necessarily the best metric to assess the residual capacity of a damaged structure. Can you please explain a bit more the basis for you statements in this paragraph?
  • In line 26 and onwards, while you statement might be correct, is there a specific reference that supports this statement? If not, you should provide sufficient data on the manuscript that can support this statement or explain in more detail how the removal time interval (I guess you mean how fast a column is removed from the analysis model) affects the structural response. In general, in this paragraph you have many unjustified statements that are not supported by any data, calculations, and/or references. Please revise accordingly.

2.3  Robustness Curves

  • It is unclear to me what exactly you mean in line 140 when you say “represent different locations along the frame of the presumed damage”. Which is the frame?
  • Also, you mention that if the steepness of the robustness curve is lower, the robustness of the structure is higher. Can you please explain this in more detail? Do you mean to say that as the area under the robustness curve is higher, the robustness is higher?
  • Can you please explain all the steps on the nice flowchart that you have on Figure 2 in a bit more detail?
  1. Blast-induced local damage for RC columns

3.1  Blast Load

  • Line 161: strictly speaking per the plot on Figure 3 the pressure goes negative not at time t0 but rather at (ta+t0). Please change the description accordingly to be consistent with the figure.

3.2 Local Models and Analyses for Blast-Damage Assessment

  • Note that for blast loaded members there are well-established response limits such as those defined on PDC-TR 06-08. Using these response limits could have been a good alternative for step (a) (line 186) for performing static non-linear analysis. I would encourage you to review the PDC document and consider implementing these well-established response limits in your work instead of using response limits that are not based on any code provisions.
  • On the equations starting in line 211, can you please add nomenclature for all the terms? There are symbols that do not have a callout.
  • In lines 217 and 233, are you really referring to Figure 4 or you wanted to refer to Figure 5?
  • On your SAP2000 analysis, how do you account for the dynamic properties of the materials? Concrete and steel are materials that are known to have higher compressive strength (for concrete) and higher yield strength (for steel) when loaded dynamically. Also, can you please provide the nominal material properties of the concrete and rebar that the RC column? I see that you provide them later, in line 264, but you should at least refer to this line when you first talk about the local analysis model of the column.
  • It is not clear how for a failed element the displacement drops to zero. Can you please elaborate more about this one?
  • Also, the standoff/weight combinations that you use are fine. However, it is common that a more representative parameter of blast intensity is the scaled standoff distance [Z=R/(W)^1/3], as you have it on eq (5).) I would suggest adding the scaled standoff on the legends of Figure 5. Also, the discussion about the sensitivity to stand-off distance and charge weight is kind of redundant. That is because the scaled standoff equation answers this question without the need to perform any analyses. The smaller the value of scaled standoff, the higher the intensity of the blast load.
  • In addition to the previous comment, because there are different pairs of standoff distance and charge weight that have the same effect on the column, a more general parameter is the scale standoff instead of pairs of standoff and charge weight. Using the scaled standoff, the plot on Figure 8 will become more general. At its current state, it is unknown what is the dmax for other combinations of standoff and charge weight. This is something that you should consider revising.
  1. Application to an Existing Structure

4.1 Case Study Structure and FEM Model

  • Is the plot on Figure 10 referring to column removal at what location?

4.2 Global Robustness Results and Sensitivity Analysis

  • As I previously mentioned, it is my understanding that you quantify robustness based on the residual lateral capacity of the structure. How do you evaluate robustness in terms of the residual capacity for gravity loads, especially at the vicinity of the removed column(s)? This is something that is missing from the discussion and your methodology. For example, a damaged structure might be still relatively strong for lateral loads but weak for gravity loads.
  • Line 302: It is important to explain clearly that what you call “damage level” on the global robustness curves is the number of columns that are removed. Instead of having this clarification in parenthesis, I suggest that you explain what the term “damage level” means in the context of the global frame analysis.
  • Line 353: it is not accurate to say “arrival time” for the peak element displacement. Might be better to say something like “the time at which the element reaches its peak displacement”. Also, on the term td of Figure 8, do you include the arrival time of the blast pressure plus the time at which the element reaches its peak displacement?
  • Line 362: how do you conclude in your analysis that there is progressive collapse of the structure? Can you perhaps add some views of your analysis models that are representative of Damage Level 0.5, 1, 1.5, 2 that are consistent with the points that you have on Figure 14?

4.3 Blast-Scenario Dependent Robustness

A general recommendation is to describe in more detail (if possible) these steps. These are very important steps, and they deserve more attention so that readers can understand your proposed methodology.

Editorial Comments

Overall, the format of the paper is nice, and the information is easy to follow. The figures are nice and look good overall. However, the manuscript has a number of spelling and grammar errors. Below is a list for a few of them but there might be more. The list below has only a few of them and I strongly recommend to carefully proofread the entire manuscript and edit as needed.

  • Line 34-35: “The interest in the explosion” does not make a lot of sense. Consider revise to something like “interest in blast-induced damage started after an important event….”
  • Line 40: Consider changing “despite to this” with “however”.
  • Line 40: Change “was able to furnish” with “furnished”.
  • Line 78: Replace “together” with “along”.
  • Line 80: Replace “matches” with “is consistent”.
  • Line 85: “allow an over strength” does not make a lot of sense (I understand what you are trying to say). Rephrase so that it is grammatically correct.
  • Line 89: Delete “a” after “which”.
  • Line 90: This sentence does not make a lot of sense – “nonlinear behavior structural materials is considered”. Please rephrase.
  • Line 99: “load-bearing” not “loadbearing”.
  • Line 100: delete “the”.
  • Line 102: “the column removal is usually applied” is wrong. Perhaps it could be “the column is removed under quasi-static loading/unloading conditions, thereby not capturing the dynamic effects of a sudden column loss”. This is just a suggestion on how to perhaps change this sentence. But the sentence as is, it is incorrect.
  • Line 107: “consists of” not “consists in”.
  • Line 134: “above-mentioned” can be replaced with “aforementioned”
  • Line 138: The y-axis cannot report. It can perhaps represent something.
  • Line 204: replace “is” with “are”.
  • Line 215: instead of “ignition point” you could perhaps say “detonation point”.
  • Line 272: Did you mean to say “extracted” instead of “extrapolated”?

Reviewer 2 Report

This paper deals with the progressive collapse of reinforced concrete frames under extreme blast load scenarios. A quantitative procedure aimed at evaluate the structural robustness is presented and applied to a 2D RC frame structure. Main original contribution is the “damage-presumption approach” adopted by the authors to obtain the “Blast-scenario dependent robustness curves”.

The paper deals with an interesting and important problem in structural mechanics. The manuscript is well written, and the first part (sections 2 and 3) is a nice introduction to the main aspects of the global robustness of RC frames. The application treated in section 4 is complete and describes the theoretical aspects in the right detail.

Overall, the article has the scientific merits and originality to be published on Vibration, in my opinion. In the sequel I collect just a few minor remarks that the authors can consider in preparing the final version of their work (I don't need to check the revised version of this article).

Line 233: Figure 4 should be changed in Figure 5.

Figure 6: the length of the column should be specified.

Line 247: Figure 7 should be changed in Figure 8.

Line 270: Figure 5 should be changed in Figure 6.

Line 271: geometric.

Line 272: from.

Line 272-279: The description of the membrane effect is not clear to me and should be better explained, in my opinion (see also the “catenary effect” in line 288).

Line 337: Please, check the English.

Line 408: maximum.

Line 415-435: Please, check the references to the figures. For example, sub-figures 20a-d are not present.

Figure 18-24: Please, indicate the (Delta t) unit of measure.

References 7-8: reference 8 should be Part II (check also volume number and page number).
